# New Insights into GFAP Negative Astrocytes in Calbindin D28k Immunoreactive Astrocytes

**DOI:** 10.3390/brainsci8080143

**Published:** 2018-08-03

**Authors:** Jie Xu

**Affiliations:** 1Department of Cellular Physiology (in Biomedical Centre Munich—BMC), Ludwig-Maximilians-University Munich, 82152 Planegg-Martinsried, Germany; xu@wickerklinik.de; Tel.: +49-6172-103101; 2Wicker Clinic, 61348 Bad Homburg, Germany

**Keywords:** glial fibrillary acidic protein (GFAP), Calbindin D28k, cultured astrocytes

## Abstract

Glial fibrillary acidic protein (GFAP) is commonly used as a specific marker for the identification of astrocytes. Nevertheless, it is known from the literature that astrocytes in situ in contrast to cultured astrocytes may feature lower levels of GFAP. In order to characterize the properties of GFAP in Calbindin D28k immunoreactive astrocytes, we use primary astrocyte cultures from cells of new-born mice. A double fluorescence immunocytochemical analysis reveals that GFAP in cultured Calbindin D28k astrocytes behaves differently depending on whether the medium contains foetal bovine serum (FBS) or not. The novelty in our study is, however, that a high percentage of Calbindin D28k cultured astrocytes in a medium with 10% FBS are GFAP negative. In addition, the study shows that Calbindin D28k astrocytes have (i) a different morphology and (ii) a higher concentration of Calbindin D28k in the nucleus than in the cytoplasm. The study provides new evidence that in order to fully understand the characteristics of astrocytes, astrocytes which are Calbindin D28k positive have to be investigated.

## 1. Introduction

Astrocytes are the major glial cell of the CNS. It’s assumed in the literature that astrocytes contain characteristic intermediate filaments, called glial filaments, which are mostly polymers of glial fibrillary acidic protein (GFAP). Therefore, GFAP can be used for astrocyte identification [1]. Put differently, the immunocytochemistry of GFAP is crucial for the identification of astrocytes [2]. Astrocytes are morphologically subdivided into protoplasmic astrocytes with short, thick, branching protrusions in the grey matter of the CNS and fibrous astrocytes with numerous, long and thin protrusions in the white matter of the CNS. The morphological differences are due to the cell content of GFAP [3].

However, there is a controversy surrounding the fact that the marker GFAP serves to identify astrocytes. Firstly, there is research arguing that the amount of GFAP in astrocytes can vary. Numerous studies have demonstrated an increased amount of GFAP in reactive astrocytes in different types of lesions or in cultured astrocytes [4,5,6]. Secondly, it has become clear that not all astrocytes in vivo express GFAP, or only weakly so [6,7,8,9,10]. These studies have revealed a remarkable heterogeneity among astrocytes, the elucidation of which is ongoing [11,12]. There is even evidence in the literature that subpopulations of astrocytes are GFAP-negative [13,14,15]. The findings question whether the expression of GFAP is at all comparable in different astrocytes.

In the present study, a double fluorescence immunocytochemical analysis was used to examine the co-localization of Calbindin D28k and astrocytic marker GFAP in the cultured astrocytes from mice. We have made a novel observation about the GFAP modification in Calbindin D28k astrocytes under various conditions. In a medium with 10% FBS, the amount of GFAP gradually decreases, while in a medium without FBS, the amount of GFAP stays almost the same. These results could be support for the studying the characteristics of GFAP and a functional analysis of Calbindin D28k immunoreactive astrocytes.

## 2. Materials and Methods

### 2.1. Preparation of Cell Cultures

Cortical astrocytes were prepared from new-born mice (1 to 3-day-old mice were used). For organ harvesting, wild-type mice are used (C57BL/6, Calbtm 1–15.B, 2.B, 1.I, Frox 26.B and GtRosa 26 tm1.I breeding mice, own breed from animal house of the physiological institute at the Ludwig-Maximilians-University in Munich).

Two tissue flasks were coated with 10 μg/mL of poly-l-Lysine overnight at 4 °C, rinsed with water, and dried for 15 min.

Hemispheres from the whole forebrain of mice were separated. Meninges and capillary vessels were removed. The tissues were cut into small pieces and incubated for 30 min. at 37 °C in a Hanks-balanced saline solution (HBSS) in which 2 mL contain 200 μL, 0.25% trypsin, and 20 μL, 1 mg/mL DNAse. The medium containing 10% foetal bovine serum (FBS) was then added to the dissociation reaction. It was removed once the cortex tissue pieces had sunk to the bottom of the tube, and replaced with a new medium. The cortex tissue pieces were mechanically dissociated using a 10 mL plastic pipette and subsequently with a flamed glass Pasteur pipette. In order to remove the fibroblasts, the cells suspension was pre-adhered in an uncoated 10 cm cell culture vessel for 30 min. at 37 °C. The supernatant was removed and split into two poly-l-lysine coated flasks. The cells were cultured in the medium containing 10% FBS for 4 days and the medium was changed the following day.

On day 4, the cells were passaged for the first time. The cells were incubated with 1 mL of 0.25% trypsin-EDTA in the cell incubator for 5 min. After the trypsin neutralization with a 20 mL medium containing 10% FBS, the cells were centrifuged for 5 min. at 1000× *g* (no brake). The supernatant was removed and the cells resuspended in 1 mL of medium containing 10% FBS. The number of cells was determined using a hemocytometer. The cells (20,000 cells/50 μL) were plated with poly-l-Lysine coated coverlids for histochemical staining. They were cultured for 9 days in a medium which either contains or did not contain 10% FBS. The medium was changed the following day in order to remove the FBS.

### 2.2. Immunocytochemistry

Astrocytes cultured on coverlids were fixed with 4% PFA (Paraformaldehyde) for approx. 20 min. and permeabilized with a permeabilization buffer (PBS-CMF, 10% BSA and 0.1% Triton), which contains 0.1% Triton X-100 for 20 min. at room temperature. Non-specific binding was blocked by incubation with a block buffer with 10% BSA (Bovine Serum Albumin) for 20 min. at room temperature. The cells were incubated with primary antibodies in an antibody dilution buffer with 1% BSA and 0.1% Triton X-100 overnight at 4 °C. The following day the cells were washed with PBS and incubated with Alexa Fluor-conjugated secondary antibodies (1:2.000) and DAPI for nuclear staining in antibody dilution buffer with 1% BSA and 0.1% Triton X-100 at room temperature for 1 h, washed with PBS and mounted on glass slides using a mounting medium (Mowiol). Appropriate primary and secondary antibodies were used in double labeling. All secondary antibodies were tested for cross-reactivity and nonspecific reactivity. The primary antibodies used here as follows: anti-GFAP (1:1000; ab53554, Abcam, Cambridge, UK), anti-Calbindin D28k (1:500, SWANT), anti-S-100 (1:100, ab4066, Abcam), anti-ß-Tubulin III (1:1000, ab7751, Abcam); and the secondary antibodies are as follows: Alexa Flour 488 Donkey Anti-Rabbit, Alexa Flour 594 Donkey Anti-Goat, and Alexa Flour 594 Goat Anti-Mouse (Invitrogen, Carlsbad, CA, USA). Images were captured using a laser scanning confocal microscope (Zeiss LSM 710, Carl Zeiss GmbH, Germany) or Epifluorescence microscopy Zeiss S100.

### 2.3. Counts

The number of Calbindin D28k positive glial cells on every coverlid was counted. The percentage of Calbindin D28k and GFAP positive glial cells was calculated by dividing the number of positive cells by the total number of Calbindin D28k glial cells per coverlid. The total number of cells (or rather cell nuclei) on every coverlid was also counted with DAPI. The percentage of Calbindin D28k positive glial cells was calculated by dividing the number of positive cells by the total number of cells (or rather cell nuclei) per coverlid. Every counting was repeated three times.

A *t*-test for a two-sided distribution and unpaired data is carried out for statistic evaluation. Differences are considered significant from a *p*-value of <0.05.

## 3. Results

We performed a time course study of the expression of GFAP in Calbindin D28k positive glial cells. Just one day after the passage, the Calbindin D28k positive glial cells are visible.

In a medium which contains FBS, the majority of Calbindin D28k glial cells are stained positive for GFAP at day 1, while after 9 days only a small percentage of Calbindin D28k glial cells are stained positive for GFAP (Figure 1, Figure 2 and Figure 3). According to our figure, it seems that Calbindin D28k was localized mostly in the nucleus and GFAP was localized in the cellular processes. Thus, due to their expression location, it does not show an overlapping colour response.

The number of Calbindin D28k positive glial cells from day 1 to day 9 in the medium with 10% FBS increased from 23 to −76 to −143 to −136 and finally to 181. The percentage of Calbindin D28k positive glial cells from day 1 to day 9 in the medium with 10% FBS increased from 1.85 × 10^−3^ to −3.35 × 10^−3^ to −3.78 × 10^−3^ to −3.47 × 10^−3^ to −4.28 × 10^−3^ (Figure 4). But the percentage of GFAP positive cells from day 1 to day 9 in the medium with 10% FBS gradually decreased from 64.69% to −36.54% to −32.07% to −23.38% and finally to −17.47% of the total number of Calbindin D28k positive glial cells. This result can be confirmed via a computer-based calculation (Curve expert 1.4). At the outset of the investigation, 92% of the Calbindin D28k positive glial cells are GFAP positive (Figure 5 and Figure 6). All Calbindin D28k positive glial cells are S100ß positive, but negative with fibronectin, CD31, aquaporin 4, and antibodies against oligodendrocyte (data not shown). In other words, all Calbindin D28k positive glial cells are astrocytes.

The number of Calbindin D28k positive glial cells from day 1 to day 9 in the medium without FBS increased from 21 to −25 to −40 to −41 and finally to 82. The percentage of Calbindin D28k positive glial cells from day 1 to day 9 in the medium without FBS firstly increased from 3.89 × 10^−3^ to −4.87 × 10^−3^ to −6.47 × 10^−3^, then decreased to −4.74 × 10^−3^ to −4.25 × 10^−3^ (Figure 4). However, in the medium without FBS, the percentage of GFAP positive cells from day 1 to day 9 stayed almost the same: from 63.81% to −52.67% to −53.40% to −62.33% and finally to −55.20% of the total number of Calbindin D28k positive glial cells. As already explained in the methods, approximately 24 h after the passage, the medium with 10% FBS is replaced by a medium without FBS. Interestingly, in the absence of FBS in the culture medium, there are noticeably higher, but nearly unchanged levels of GFAP in Calbindin D28k positive glial cells. No further reduction of the percentage of GFAP positive glial cells can be seen (Figure 5). In addition, the Calbindin D28k occurs in astrocytes in the nucleus with a much higher concentration than in the cytoplasm not only in the medium with 10% FBS but also in the medium without FBS.

It was also observed that in the absence of FBS a differentiation was taking place. The increase in Calbindin D28k positive astrocytes was accompanied by morphological changes. They develop protrusions, their polygonal form changed into a star-shaped form (Figure 7, Figure 8, Figure 9, Figure 10, Figure 11, Figure 12, Figure 13 and Figure 14).

It was also observed that several neural extensions of Calbindin D28k expressing neurons are in close contact with Calbindin D28k astrocytes. This shows a tight relationship between Calbindin D28k astrocytes and nerve cells which is advantageous for neurons (Figure 15 and Figure 16).

## 4. Discussion

In this study, we focus on cultured Calbindin D28k immunoreactive astrocytes which are GFAP negative, not positive as standardly assumed in the literature [16,17]. The observations allow new insights into the behaviour of GFAP in cultured astrocytes under different conditions. The study shows that only the investigation of GFAP negative astrocytes leads to the complete understanding of the role of GFAP as a marker for astrocytes.

As already mentioned in the introduction, GFAP is standardly used as a marker for astrocytes, especially for cultured astrocytes [18,19,20]. This is based on results such as those reported in Du et al. [19], who found that in a primary culture of astrocytes taken from new-born mice 90% are GFAP positive. The present study doubts that GFAP is a reliable marker for astrocytes. It shows that a substantial amount of cultured Calbindin D28k astrocytes are GFAP negative. The percentage of GFAP negative cells from day 1 to day 9 in the medium with 10% FBS gradually increased from 35.31% to −63.46% to −67.3% to −76.62% and finally to −82.53% of the total number of Calbindin D28k immunoreactive astrocytes. Therefore, analyses using GFAP as a marker may lead to wrong results.

Secondly, the results of the present study also provide further evidence against the assumption that GFAP negative astrocytes are only a subpopulation within a culture of astrocytes which are generally GFAP positive. We observe that the expression of GFAP in Calbindin D28k immunoreactive astrocytes depends on conditions in the astrocytes’ environment. We find that the presence vs. absence of FBS in the medium plays an important role. In a medium, which contains 10% FBS, GFAP gets downregulated after passage. This is in contrast to a medium without FBS, where the amount of GFAP stays almost the same. In other words, astrocytes are argued to be heterogeneous [21]. Walz and Lang [13] observed subpopulations without GFAP [10]. In the present study, there is, however, no evidence for such a heterogeneity of astrocytes. Calbindin D28k immunoreactive astrocytes at day 9 in a medium with 10% FBS are generally GFAP negative because it does not get expressed. With the computer-based calculation, all Calbindin D28k astrocytes are presumably GFAP positive at the beginning of a passage in a medium with 10% FBS. This study shows that the expression of GFAP in Calbindin D28k immunoreactive astrocytes is dynamic.

However, the percentage of GFAP positive cells from day 1 to day 9 in the medium without serum remains at approximately the same level as that of the total number of Calbindin D28k astrocytes, although GFAP is degraded. It can be assumed that all newly formed Calbindin D28k immunoreactive astrocytes are daughter cells of the original cells and in this way the proportion (GFAP) of approx. 60% of Calbindin D28k astrocytes can be maintained. If the spontaneous new formation of Calbindin D28k immunoreactive astrocytes were to occur, the percentage of GFAP positive cells of Calbindin D28k astrocytes would have to change.

Further, the study at hand shows that the presence vs. absence of serum has an impact on the amount of GFAP expressed in astrocytes containing Calbindin D28k. Interestingly, the lack of serum leads to important changes. Firstly, the amount of GFAP increases significantly, that is, the astrocytes become reactive. This result is unexpected because, in the literature, the withdrawal of serum and the starvation of serum is reported to cause a decrease in GFAP mRNA levels. Only astrocytes in a serum with rich growth factors are found to be reactive and expressing higher levels of GFAP [22]. Secondly, the lack of serum in the medium leads to a change in the morphology of the Calbindin D28k immunoreactive astrocytes, more specifically, from a polygonal occurrence to a gigantic star-like appearance. Kettenmann and Verkhratsky [20] pointed out that, an archetypal morphological feature of astrocytes is their expression of intermediate filaments, which form the cytoskeleton. The main type of astroglial intermediate filament protein is GFAP. Astrocytes can considerably increase their volume, particularly under pathological conditions. The stretch-activated Ca^2+^-permeable channels could link the regulation of cell volume to intracellular Ca^2+^ signalling [23]. Calbindin D28k belongs to the large family of EF-hand calcium-binding proteins. Structurally these proteins are characterized by the presence of a variable number of evolutionary well-conserved helix-loop-helix motives, which bind Ca^2+^ ions with high affinity. Functionally, they fall into two groups: by interaction with target proteins, calcium sensors translate calcium concentrations into signalling cascades, whereas calcium buffers are thought to modify the spatiotemporal aspects of calcium transients [24]. In view of the documents described above, Calbindin D28k immunoreactive astrocytes with more GFAP may fulfil several functions and be able to adapt to the environment without FBS.

Thirdly, cultured Calbindin D28k astrocytes are special with respect to the intensity of their Calbindin D28k immunoreactivity. While it is less intense in the cytoplasm it is more intense in the nucleus. It is well known that the synthesis of proteins takes place in the cytoplasm. Christie et al. [25] invoked that the translation of proteins can only take place in the cytoplasm. However, they came to the conclusion that many proteins are only able to fulfil their function then, if they are in the nucleus. The active uptake of proteins into the nucleus is a fast, specific and evolutionary conserved process [26,27]. The spatial separation of transcription and translation provides eukaryotes with powerful mechanisms for controlling gene expression [28]. Regulatory proteins, such as RNA binding proteins, play an important role throughout gene expression from transcription to translation [29,30]. In our study, we observe whether the presence vs. absence of FBS has an impact on the intense of Calbindin D28k in the nucleus. We arrive at the conclusion that the more intense Calbindin D28k in the nucleus does not change. It is as high in a medium with 10% FBS as in one without FBS. This finding could be linked to the fact that Calbindin D28K in the nucleus is responsible for many cell functions and therefore does not get downregulated. Calbindin D28K in the nucleus possibly regulates the gene expression in the cell nucleus and thus influences all cell functions. To investigate the underlying mechanisms leading to and the consequences of the more intense Calbindin D28k in the nucleus would be important questions for future research on Calbindin D28k astrocytes.

In this study, we find that Calbindin D28k astrocytes influence neurons in their neighbourhood. Several neuronal processes next to Calbindin D28k astrocytes seem to develop thicker and stronger than others without contact with Calbindin D28k astrocytes. This raises questions about signalling processes between neurons and astrocytes for the benefit of neurons. Although we cannot answer the question how neurons and astrocytes communicate with each other, we suspect that Calbindin D28k astrocytes are able to produce a still unknown factor, which has positive effects on the development of neurons close to them. It has already been reported in the literature that astrocytes are able to secrete a whole series of neurotrophic proteins, e.g., NGF, BDNF, NT-3, bFGF CNTF, as well as neuropeptides, somatostatin and substance P, which can further promote the survival of neurons and regrowth [12,16,31,32].

Another important result of the present study is the decrease in Calbindin D28k immunoreactive astrocytes in the medium without serum following the observed increase. The assumption is again that the Calbindin D28k astrocytes produce an unknown factor that improves the condition of the cell. It can further promote the survival and regrowth of neurons. This improved condition leads to the observed decrease in Calbindin D28k astrocytes. Under the improved conditions, the cells appear to require less Calbindin D28k and therefore to experiment less.

## 5. Conclusions

In this study, we do research the GFAP negative astrocytes in cultured Calbindin D28k immunoreactive astrocytes. The observations make it possible to reveal an aspect of GFAP in cultured astrocytes under various conditions which until now has been underappreciated. Our findings provided clear evidence of the biological characteristics of Calbindin D28k astrocytes. However, it needs to be stated that there is much more research to be done. One possible research question would be to investigate the elevated concentration of Calbindin D28k in the nucleus of Calbindin D28k astrocytes. The focus could be on the factors which may influence the growth of the surrounding neurons.

## Figures and Tables

**Figure 1 brainsci-08-00143-f001:**
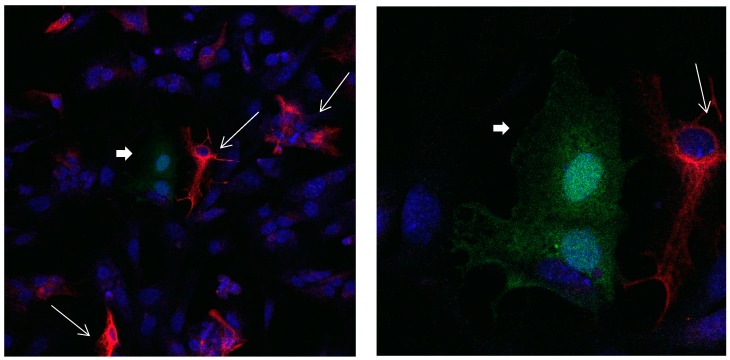
Cultures in a serum-containing medium, 1 day after passage: Immunofluorescence with anti-GFAP (glial fibrillary acidic protein) (red, secondary antibody: donkey anti-goat IgG Alexa Fluor 594), and with anti-Calbindin D28k (polyclonal antibody from the company SWANT) (green, secondary antibody: donkey anti-rabbit IgG Alexa Fluor 488); Nuclear staining with DAPI (4′,6-diamidino-2-phenylindole dihydrochloride) (blue). Confocal microscopy. **Left**: zoom = 1, numerous GFAP-positive astrocytes (long arrows), in between 2 Calbindin D28k positive and GFAP-negative astrocytes (short arrows). **Right**: zoom = 4, higher magnification of the 2 Calbindin D28k positive astrocytes (short arrows).

**Figure 2 brainsci-08-00143-f002:**
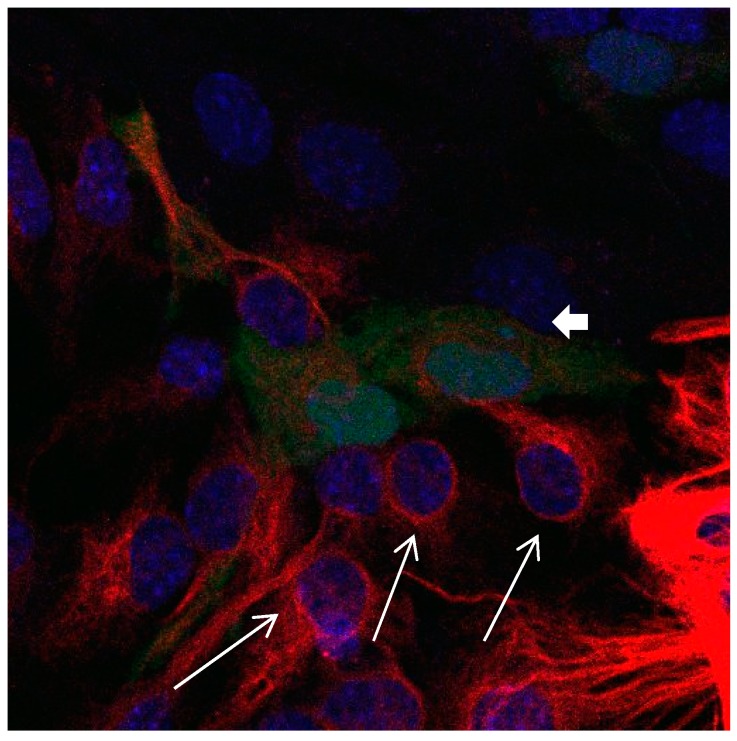
Cultures in a serum-containing medium, 5 days after passage: Immunofluorescence with anti-GFAP (glial fibrillary acidic protein) (red, secondary antibody: donkey anti-goat IgG Alexa Fluor 594), and with anti-Calbindin D28k (polyclonal antibody from the company SWANT) (green, secondary antibody: donkey anti-rabbit IgG Alexa Fluor 488); Nuclear staining with DAPI (4′,6-diamidino-2-phenylindole dihydrochloride) (blue). Confocal microscopy. Zoom = 4, numerous GFAP-positive and Calbindin D28k negative astrocytes (long arrows), in between 2 Calbindin D28k positive and GFAP-positive astrocytes (short arrows).

**Figure 3 brainsci-08-00143-f003:**
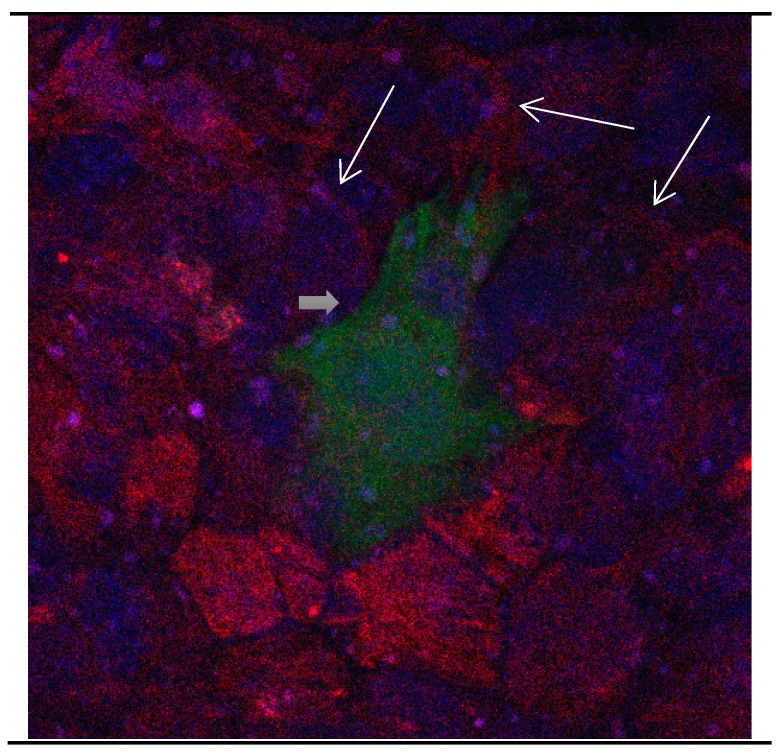
Cultures in a serum-containing medium, 9 days after passage: Immunofluorescence with anti-GFAP (glial fibrillary acidic protein) (red, secondary antibody: donkey anti-goat IgG Alexa Fluor 594), and with anti-Calbindin D28k (polyclonal antibody from the company SWANT) (green, secondary antibody: donkey anti-rabbit IgG Alexa Fluor 488); Nuclear staining with DAPI (4′,6-diamidino-2-phenylindole dihydrochloride) (blue). Confocal microscopy. Zoom = 4, numerous GFAP-positive and Calbindin D28k negative astrocytes (long arrows), in between 1 Calbindin D28k positive and GFAP-negative astrocyte (short arrows).

**Figure 4 brainsci-08-00143-f004:**
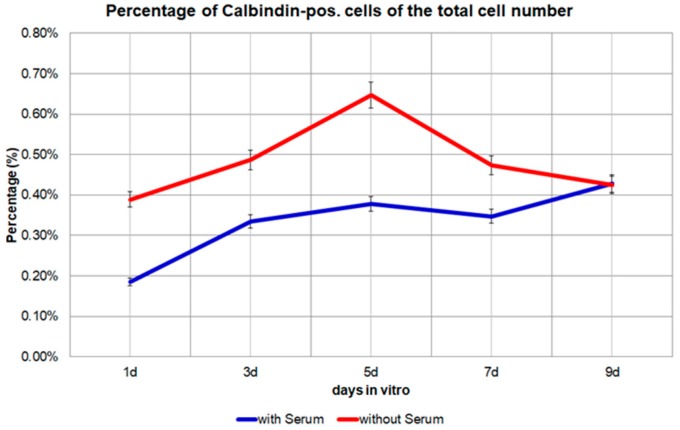
The percentage of Calbindin D28k positive cells of the total cell number from day 1 to day 9 in the medium with 10% FBS or in the medium without FBS. *p* = 0.0187, significant.

**Figure 5 brainsci-08-00143-f005:**
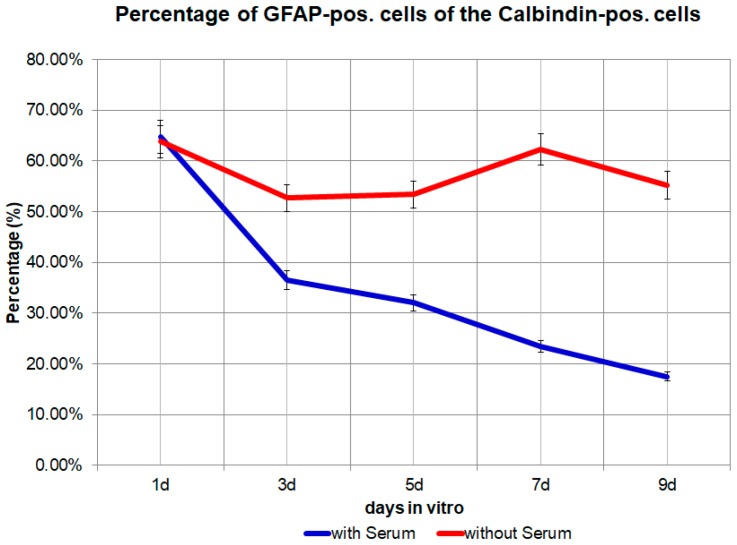
The percentage of GFAP (glial fibrillary acidic protein) positive cells from day 1 to day 9 in the medium with 10% FBS (fetal bovine serum) or in the medium without FBS. *p* = 0.0142, significant.

**Figure 6 brainsci-08-00143-f006:**
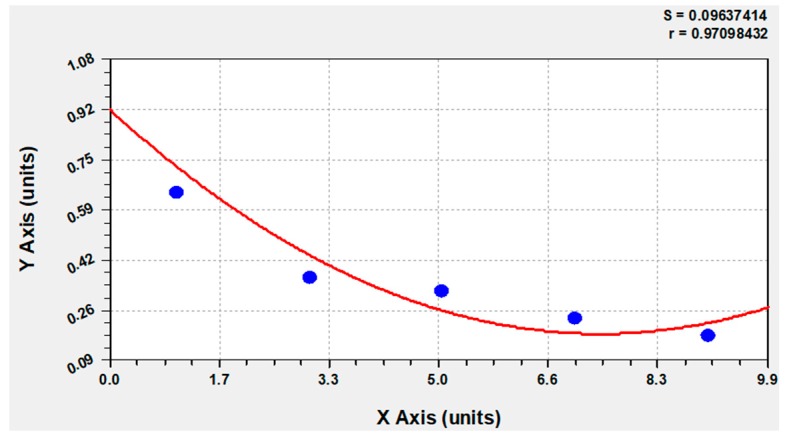
The computer-based calculation showed at the outset of the investigation 92% that Calbindin D28k positive glial cells are also GFAP-positive. The blue circle: the percentage of GFAP positive cells from day 1 to day 9 in the medium with 10% FBS gradually decreased from 64.69% to −36.54% to −32.07% to −23.38% and finally to −17.47% of the total number of Calbindin D28k positive glial cells. The red line: adjustment of the data with Curve expert 1.4.

**Figure 7 brainsci-08-00143-f007:**
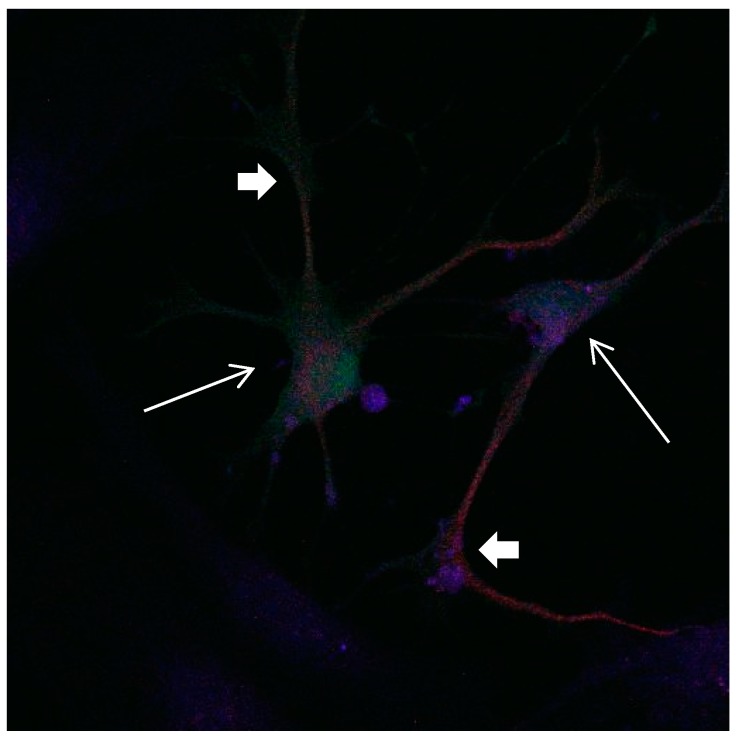
Cultures in a serum-free medium, 5 days (6 days after the passage): Immunofluorescence with anti-GFAP (glial fibrillary acidic protein) (red, secondary antibody: donkey anti-goat IgG Alexa Fluor 594), and with anti-Calbindin D28k (polyclonal antibody from the company SWANT) (green, secondary antibody: donkey anti-rabbit IgG Alexa Fluor 488); Nuclear staining with DAPI (4′,6-diamidino-2-phenylindole dihydrochloride) (blue). Confocal microscopy. Zoom = 4, 2 Calbindin D28k positive and GFAP-positive astrocytes (long arrows) with processes (short arrows).

**Figure 8 brainsci-08-00143-f008:**
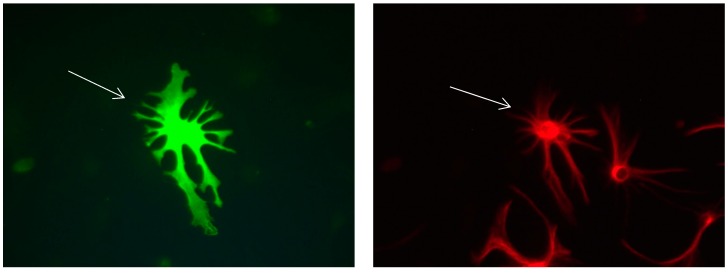
Cultures in a serum-free medium, 5 days (6 days after the passage): Immunofluorescence with anti-GFAP (glial fibrillary acidic protein) (red, secondary antibody: donkey anti-goat IgG Alexa Fluor 594), and with anti-Calbindin D28k (polyclonal antibody from the company SWANT) (green, secondary antibody: donkey anti-rabbit IgG Alexa Fluor 488). Epifluorescence microscopy. One giant Calbindin D28k positive and GFAP-positive astrocyte (long arrows) with processes.

**Figure 9 brainsci-08-00143-f009:**
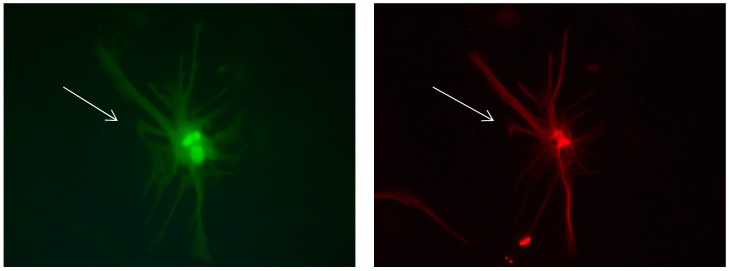
Cultures in a serum-free medium, 5 days (6 after the passage): Immunofluorescence with anti-GFAP (glial fibrillary acidic protein) (red, secondary antibody: donkey anti-goat IgG Alexa Fluor 594), and with anti-Calbindin D28k (polyclonal antibody from the company SWANT) (green, secondary antibody: donkey anti-rabbit IgG Alexa Fluor 488). Epifluorescence microscopy. One giant Calbindin D28k positive and GFAP-positive astrocyte (long arrows) with processes.

**Figure 10 brainsci-08-00143-f010:**
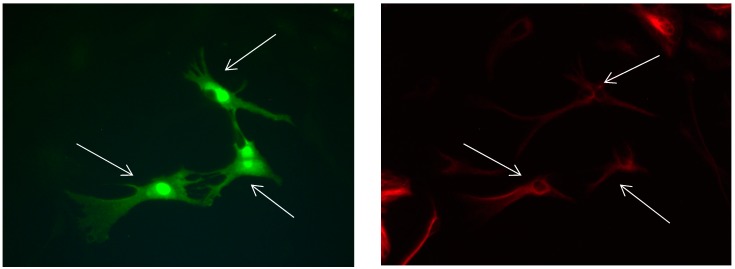
Cultures in a serum-free medium, 5 days (6 days after the passage): Immunofluorescence with anti-GFAP (glial fibrillary acidic protein) (red, secondary antibody: donkey anti-goat IgG Alexa Fluor 594), and with anti-Calbindin D28k (polyclonal antibody from the company SWANT) (green, secondary antibody: donkey anti-rabbit IgG Alexa Fluor 488). Epifluorescence microscopy. Three Calbindin D28k positive and GFAP-positive astrocytes (long arrows) with processes.

**Figure 11 brainsci-08-00143-f011:**
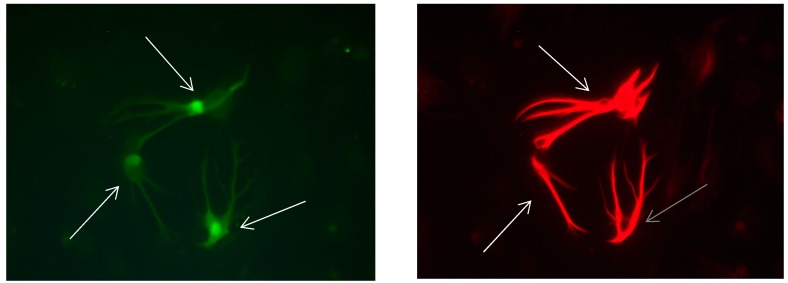
Cultures in a serum-free medium, 5 days (6 days after the passage): Immunofluorescence with anti-GFAP (glial fibrillary acidic protein) (red, secondary antibody: donkey anti-goat IgG Alexa Fluor 594), and with anti-Calbindin D28k (polyclonal antibody from the company SWANT) (green, secondary antibody: donkey anti-rabbit IgG Alexa Fluor 488). Epifluorescence microscopy. Three Calbindin D28k positive and GFAP-positive astrocytes (long arrows) with processes.

**Figure 12 brainsci-08-00143-f012:**
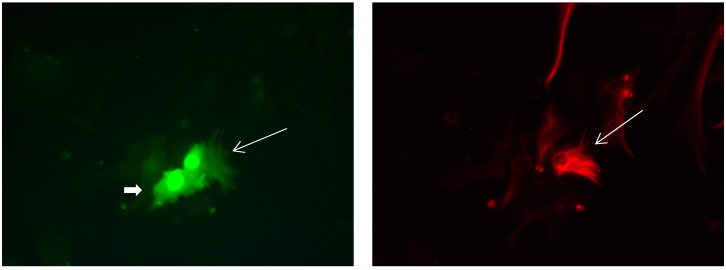
Cultures in a serum-free medium, 3 days (4 days after the passage): Immunofluorescence with anti-GFAP (glial fibrillary acidic protein) (red, secondary antibody: donkey anti-goat IgG Alexa Fluor 594), and with anti-Calbindin D28k (polyclonal antibody from the company SWANT) (green, secondary antibody: donkey anti-rabbit IgG Alexa Fluor 488). Epifluorescence microscopy. One Calbindin D28k positive and GFAP-positive astrocyte (long arrows), besides 1 Calbindin D28k positive and GFAP-negative astrocyte (short arrows).

**Figure 13 brainsci-08-00143-f013:**
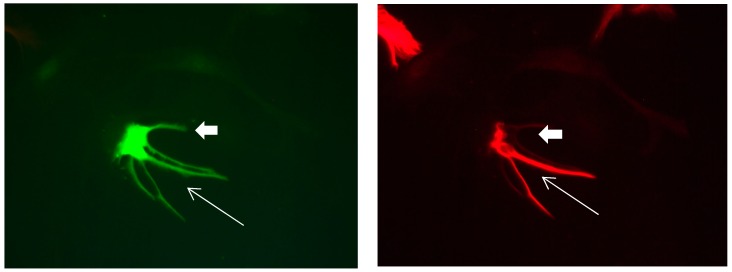
Cultures in a serum-free medium, 3 days (4 days after the passage): Immunofluorescence with anti-GFAP (glial fibrillary acidic protein) (red, secondary antibody: donkey anti-goat IgG Alexa Fluor 594), and with anti-Calbindin D28k (polyclonal antibody from the company SWANT) (green, secondary antibody: donkey anti-rabbit IgG Alexa Fluor 488). Epifluorescence microscopy. One Calbindin D28k positive and GFAP-positive astrocyte (long arrows), besides, 1 Calbindin D28k positive and GFAP-weak-positive astrocyte (short arrows).

**Figure 14 brainsci-08-00143-f014:**
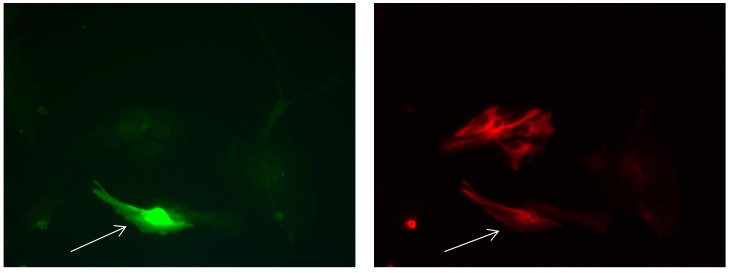
Cultures in a serum-free medium, 1 day (2 days after the passage): Immunofluorescence with anti-GFAP (glial fibrillary acidic protein) (red, secondary antibody: donkey anti-goat IgG Alexa Fluor 594), and with anti-Calbindin D28k (polyclonal antibody from the company SWANT) (green, secondary antibody: donkey anti-rabbit IgG Alexa Fluor 488). Epifluorescence microscopy. One Calbindin D28k positive and GFAP-positive astrocyte (long arrows).

**Figure 15 brainsci-08-00143-f015:**
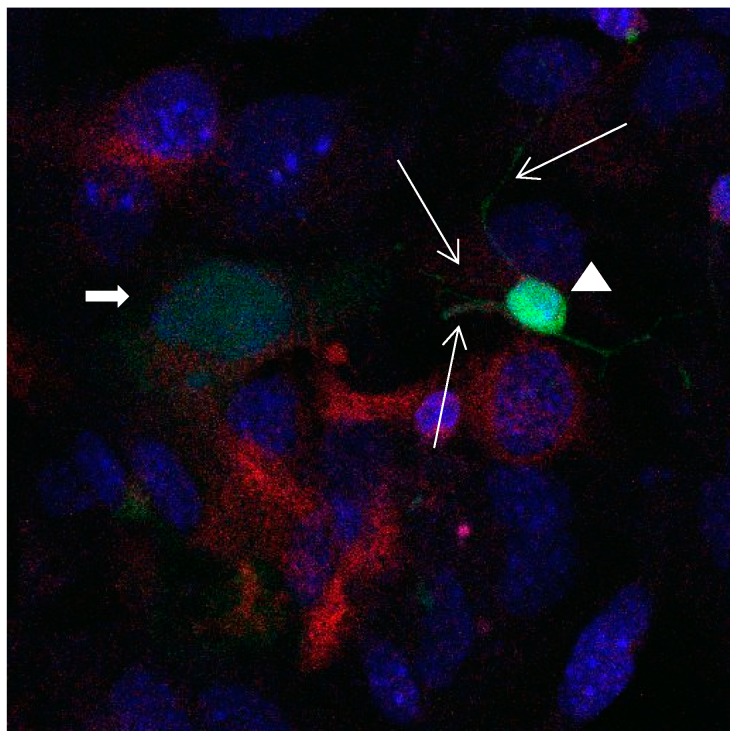
Cultures in a serum-containing medium, 1 day after the passage: Immunofluorescence with anti-GFAP (glial fibrillary acidic protein) (red, secondary antibody: donkey anti-goat IgG Alexa Fluor 594), and with anti-Calbindin D28k (polyclonal antibody from the company SWANT) (green, secondary antibody: donkey anti-rabbit IgG Alexa Fluor 488); Nuclear staining with DAPI (4′,6-diamidino-2-phenylindole dihydrochloride) (blue). Confocal microscopy. Zoom = 4. Left: a Calbindin D28k positive and GFAP positive astrocyte (short arrows). Besides: a Calbindin D28k positive and Calbindin D28k expressing neuron (triangle) with extensions (long arrows).

**Figure 16 brainsci-08-00143-f016:**
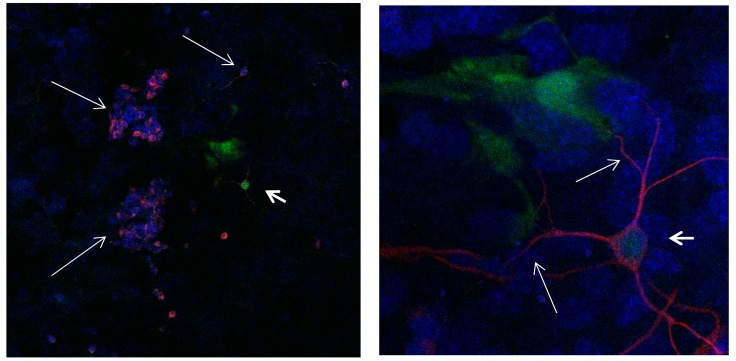
Cultures in a serum-containing medium, 5 days after the passage: Immunofluorescence with anti-ß-tubulin-III (red, secondary antibody: goat anti-mouse IgG Alexa Fluor 555), and with anti-Calbindin D28k (polyclonal antibody from the company SWANT) (green, secondary antibody: donkey anti-rabbit IgG Alexa Fluor 488); Nuclear staining with DAPI (4′,6-diamidino-2-phenylindole dihydrochloride) (blue). Confocal microscopy. **Left**: zoom = 1, numerous ß-tubulin-III-positive neurons (long arrows), on the right 1 Calbindin D28k positive and tubulin-III-positive neuron (short arrows). **Right**: zoom = 4, higher magnification of the Calbindin D28k positive and tubulin-III-positive neuron (short arrows) with extensions (long arrows). Above some Calbindin D28k positive astrocytes.

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
