# Peer review of "New Insights into GFAP Negative Astrocytes in Calbindin D28k Immunoreactive Astrocytes"

_brainsci, 2018, doi:10.3390/brainsci8080143_

Round 1

Reviewer 1 Report

Major  comments

In manuscript entitled “New Insights into GFAP negative Astrocytes in Calbindin D28k positive Astrocytes” author has tried to explore a very interesting and important topic of astrocytes characterization with GFAP and finding better method to identity astrocytes in cultures. However, the methodology and experimental design for this study is poorly executed. The major issue that I find and confused is that “on what basis author identify the presence/absence of GFAP in Calbindin D28 positive cells. My few observation regarding that are as follow:

In introduction, last paragraph, authors said have made the novel observation about the GFAP modification in Calbindin D28k astrocytes, however, they didn’t mention their findings. They should include their findings in this paragraph.

Authors have reported that most of the Calbindin D28k glial cells show GFAP staining at Day 1 but only few % of Calbindin D28K glial cells show GFAP staining at Day 9. Is there a possibility that those particular glial cells (Cal D28K) get differentiated into some other cells?

I am confused in Figure 1, authors said Calbindin D28 K glial cells show GFAP staining, but I am not seeing any co-localization of green (CalD28 cells) and astrocytes (red), if they co-localized then they should be able to see some yellow color. In fact in whole manuscript, in any figure they didn’t show any co-localization, then on what basis they are saying that CalD28 cells show/don’t show GFAP staining.

 Minor comments

There are some typos throughout the manuscript especially in abstract

Author Response

Response to Reviewer 1 Comments

In manuscript entitled “New Insights into GFAP negative Astrocytes in Calbindin D28k positive Astrocytes” author has tried to explore a very interesting and important topic of astrocytes characterization with GFAP and finding better method to identity astrocytes in cultures. However, the methodology and experimental design for this study is poorly executed. The major issue that I find and confused is that “on what basis author identify the presence/absence of GFAP in Calbindin D28 positive cells. My few observation regarding that are as follow:

Response to Reviewer 1 Comments (1): in the present study, double fluorescence immunocytochemical analysis was used to examine the co-localization of Calbindin D28k and astrocytic marker GFAP in the cultured astrocytes from mice. At every histological preparation in this manuscript, the cells were incubated with two primary antibodies (anti-GFAP and anti-Calbindin-D28k), with two Alexa Fluor-conjugated secondary antibodies (Alexa Fluor 594- red-GFAP, Alexa Fluro 488-green-Calbindin-D28k).

In Figure 2 (with short arrows), Figure 9 (with long arrows), you can clearly see two cells with co-localization of green (Calbindin-D28k positive) and red (GFAP-positive). The Imposition of light from both of markers (green and red) results in a mixed color in cytoplasmas.

In Figure 3, 4, 5, 6, 7 and 8, you can see two photos in each figure. But, it is the same photo at the same place (in another word, the same cell) with two different color channels (green: Calbindin-D28k positive, red: GFAP positive). I made it special in order to show the change of GFAP clearly, although the cells are significantly Calbindin-D28k positive.

In introduction, last paragraph, authors said have made the novel observation about the GFAP modification in Calbindin D28k astrocytes, however, they didn’t mention their findings. They should include their findings in this paragraph.

Response to Reviewer 1 Comments (2): thanks for your comments. I have already written the findings in Results and Discussion. I would like to write the findings in introduction, too.

Authors have reported that most of the Calbindin D28k glial cells show GFAP staining at Day 1 but only few % of Calbindin D28K glial cells show GFAP staining at Day 9. Is there a possibility that those particular glial cells (Cal D28K) get differentiated into some other cells?

Response to Reviewer 1 Comments (3): we have already immunostaining with antibodies against Oligodendrocyte, fibronectin, CD31, aquaporin4 performed. The Calbindin-D28k positive glia cells are negative with these antibodies. That is, oligodendrocytes, Endothelial cells, ependymal cells are already excluded. We have also immunostaining with ß-tubulin-III performed. In Figure 11, you could see a tubulin-III positive, Calbindin-D28k positive neuron. But, the Calbindin-D28k positive glia cells are tubulin-III negative.

Thank you for your comments. I could also write it in Results.

I am confused in Figure 1, authors said Calbindin D28 K glial cells show GFAP staining, but I am not seeing any co-localization of green (CalD28 cells) and astrocytes (red), if they co-localized then they should be able to see some yellow color. In fact in whole manuscript, in any figure they didn’t show any co-localization, then on what basis they are saying that CalD28 cells show/don’t show GFAP staining.

Response to Reviewer 1 Comments (4): in Figure 1, the cells were also incubated with two primary antibodies (anti-GFAP and anti-Calbindin-D28k), with two Alexa Fluor-conjugated secondary antibodies (Alexa Fluor 594- red-GFAP, Alexa Fluro 488-green-Calbindin-D28k). But, the two green cells (Calbindin-D28k positive) with short arrows are GFAP negative, of course, you are not seeing any co-localization of green (CalD28 cells) and astrocytes (red).

In Discussion, I have written: we focus on cultured Calbindin positive astrocytes which are GFAP negative. I have also written: GFAP is standardly used as a marker for astrocytes, especially for cultured astrocytes. Here I would like to show special: for cultured astrocytes, maybe GFAP is negativ.

In Response to Comments (1), I have also explained about other Figure. You could see co-localization in Figure 2, 3, 4, 5, 6, 7, 8, 9.

 Minor comments

There are some typos throughout the manuscript especially in abstract

Response to Reviewer 1 Comments (5): could you point out exactly?

Reviewer 2 Report

Introduction

The author should explain what is "calbindin D28K positive astrocyte". As far as I know, such a definition is not popular. Is it reported previously? If so, the authors should cite references. 

Astrocytic heterogeneity becomes a hot topic in the glia research. Authors should cite some recent references, e.g., John Lin et al. (2017) Nat. Neurosci. 20,  Zamanian, J. L., et al. (2012). J. Neurosci. 32, Tatsumi K.,et.al.,(2018) Front. Neuroanat.

Materials and Methods

The methods other than cell culture are not explained in detail, e.g., mice strain, breeding methods, approved protocol for animal experiments etc. should be included in this section. 

The methods how they performed quantitative analyses and statistics should be explained in more detail. 

Results

 In the Page 3, second paragraph, the author described only % of cells, but not the actual number of cells. The author should describe the cell numbers with % ratios. 

The author should show the percent ratio of the calbindin D28K positive cells to the total astrocytes (GFAP positive).

On what basis, the author says that the calbindin D28K positive/ GFAP negative cells are astrocyte? The author should define the nature of the cells with other astrocytic marker(s) such as s100beta, but the quality of the Fig.12 is too low to see the s100beta expression. This Fig should be improved. Fig.14 does not make sense to me. It should be explained in detail.

Discussion

The author frequently described that some of the cellular processes "touch" on the other type of cells. However, from the light microscopic pictures such as FIg. 11, one can never prove the "touch".  If author wants to suggest the "touch" from the morphological standpoint,  he should perform immunoelectron microscopy.

Author Response

Introduction

The author should explain what is "calbindin D28K positive astrocyte". As far as I know, such a definition is not popular. Is it reported previously? If so, the authors should cite references.

Response to Reviewer 2 Comments (1): thank you for your feedback. Maybe you think, “calbindin-D28k immunoreactive astrocytes”, is better here. But, there are two reasons, why did I write like this.

Firstly, in two studies from 1996, Toyoshima et al. and Ahmed et al. have already pointed out calbindin-D28k immunoreactive astrocytes with “calbindin-positive cells were astrocytes”, “the number of GFAP-positive reactive astrocytes was larger than that of CD28-positive ones”.

Toyoshima, T., et al (1996): Expression of calbindin-D28k by reactive astrocytes in gerbil hippocampus after ischaemia. NeuroReport 7 (13): 2087-2091.

Ahmed, B.Y., et al (1996): A chronological study of the expression of glial fibrillary acidic protein and calbindin-D28k by reactive astrocytes in the electrically lesioned rat brain. Neuroscience 26 (3): 271-278.

Secondly, my main research topic in the lab is about calbindin-D28k immunoreactive astrocytes. Colleagues reported in the literature (see above two literaure, Mattson, M.P. et al (Journal of Neuroscience Research 42: 357-370, 1995), Nancy E. J. Berman et al (Mol Chem Neuropathol 34(1): 25-38, 1998)) calbindin-D28k positive glial cells are GFAP positive, that is, calbindin-D28k positive glial cells are astrocytes. But I have often noticed: there are many calbindin-D28k immunoreactive astrocytes with GFAP negative. GFAP is here flexible and dynamic, but with rule. Please have a look on Fig. 13 and Fig. 14. The percentage of GFAP positive cells from day 1 to day 9 in the medium with 10% FBS gradually decreased from 64.69% to 17.47%. Via a computer-based calculation, at the outset of the investigation 92% calbindin-D28k cells are GFAP positive, that is approximately 100%. However, in the medium without FBS the percentage of GFAP positive cells from day 1 to day 9 stayed almost the same. Probably, this is evidence for a subpopulation of astrocytes that are calbindin-D28k immunoreactive astrocytes. More specifically “calbindin D28K positive astrocyte”. This is also the main content in another manuscript. I explain it in detail in that manuscript.

But, I would like to change it: new insights in GFAP negative astrocytes in calbindin-D28k immunoreactive astrocytes.

Astrocytic heterogeneity becomes a hot topic in the glia research. Authors should cite some recent references, e.g., John Lin et al. (2017) Nat. Neurosci. 20,  Zamanian, J. L., et al. (2012). J. Neurosci. 32, Tatsumi K.,et.al.,(2018) Front. Neuroanat.

Response to Reviewer 2 Comments (2): a hot topic, thank you very much. I could also cite more recent references.

Materials and Methods

The methods other than cell culture are not explained in detail, e.g., mice strain, breeding methods, approved protocol for animal experiments etc. should be included in this section.

Response to Reviewer 2 Comments (3): cortical astrocytes were prepared from newborn (1 to 3 days old) mice. For organ harvesting, wild-typ mice are used (animal house of the physiological institute at the Ludwig-Maximilians-University in Munich).

The different wild-typ mice were used. There is no special requirement for mice strain and breeding methods.

The methods how they performed quantitative analyses and statistics should be explained in more detail.

Response to Reviewer 2 Comments (4): The number of Calbindin D28k positive glial cells on every coverlid was counted. The percentage of Calbindin D28k and GFAP positive glial cells was calculated by dividing the number of positive cells by the total number of Calbindin D28k glial cells per coverlid. Every counting was repeated three times.

A t-test for a two-sided distribution and unpaired data is carried out for statistic evaluation. Differences are considered significant from a p-value of <0.05.

The goal of this research is to investigate the character of GFAP in calbindin-D28k immunoreactive astrocytes. The number of Calbindin cells and the number of GFAP positive Calbindin cells were counted separately. This is already in detail. About statistics is also in detail. Colleagues can repeat with this “protocol”.

Results

 In the Page 3, second paragraph, the author described only % of cells, but not the actual number of cells. The author should describe the cell numbers with % ratios.

Response to Reviewer 2 Comments (5): I would like to complete the number of cells, in another word, the number of calbindin-D28k immunoreactive astrocytes with GFAP positive / GFAP negative.

The author should show the percent ratio of the calbindin D28K positive cells to the total astrocytes (GFAP positive).

Response to Reviewer 2 Comments (6): The goal of this research is to investigate the character of GFAP in calbindin-D28k immunoreactive astrocytes. The cell co-culture is carried out in this experiment. This means, there are also oligodendrocytes, endothelial cells, ependymal cells, neurons together, except astrocytes. The percent ratio of the calbindin D28K positive cells to the total astrocytes, this is very difficult to perform. Please do not forget, “astrocytic heterogeneity becomes a hot topic”. Many colleagues have reported about GFAP-neg. astrocytes. GFAP-positive does not mean “the total astrocytes”.

On what basis, the author says that the calbindin D28K positive/ GFAP negative cells are astrocyte? The author should define the nature of the cells with other astrocytic marker(s) such as s100beta, but the quality of the Fig.12 is too low to see the s100beta expression. This Fig should be improved. Fig.14 does not make sense to me. It should be explained in detail.

Response to Reviewer 2 Comments (7): S100beta is also an important astrocytic marker. Every Calbindin-D28k immunoreactive glia cell here is s100beta positive. And every Calbindin-D28k positive neuron here is s100beta negative. I have explained Fig. 14 at the beginning (Response to Reviewer 2 Comments (1)). With both evidence, I say that the calbindin-D28k positive / GFAP negative glia cells are astrocytes.

Discussion

The author frequently described that some of the cellular processes "touch" on the other type of cells. However, from the light microscopic pictures such as FIg. 11, one can never prove the "touch".  If author wants to suggest the "touch" from the morphological standpoint,  he should perform immunoelectron microscopy.

Response to Reviewer 2 Comments (8): thank you for your feedback. But, at Fig. 10 and Fig. 11, please pay more attention to: this is the confocal microscopic picture. In contrast to conventional light microscopy, not all of the specimen is illuminated, but at any time only a fraction of it. The cellular processes of a neuron at Fig. 10 and Fig. 11 could here be described: touch on a Calbindin-D28k pos. astrocyte.

Round 2

Reviewer 1 Report

Major comments

In materials and methods author mentioned that the astrocytes cultures were passaged for the first time, so when time course study performed, it was after passaged , can author provide immunostaining data of Day1 of culture, before passaging them.

Results section, page3, first paragraph, author mentioned in manuscript that “the majority of Calbindin D28k glial cells are stained positive for GFAP at day 1, while after 9 days only a small percentage of Calbindin D28k glial cells are stained positive for GFAP (Figure 1, Figure 2)” , here figure 1 shows day 5 immunostaining data not day 9 data, can author provide immunostaining data of day 9. (in earlier version Figure 2 was showing day 5 data, any reason of changing figure numbers?)  

Morphology of astrocytes look totally different in cultures of serum –free medium at day 5 (Figure 3), can author provide immunostaining data of culture at 24h in serum-free medium to make comparison.

For Figure 1, can author provide single stained (green and red channel) images as with merged image shown in figure 1, I only see two cells positive with Calbindin D28, and those are not GFAP positive  as claimed in results (1st paragraph)

For figure 3,4, 5, 6, 7 and 8, can author provide merged and phase contrast images ?

Discussion, paragraph 3, author didn’t do any analysis on GFAP expression in 10% FBS cultures, can author provide image analysis or western blot data to claim that GFAP is downregulating after passage?

Minor comments:

Examples of typos and grammar mistakes

Abstract, line 6, “differ-ently”

Abstract, line 7, “nov-elty”

Abstract, line 2nd to last , as-trocytes

Abstract, line 2, “Nevertheless, it known”, please correct the grammar.

Page 2, material and methods section, 2nd paragraph, line 3, “ 20 ul 1 mg/ml DNAse”, “coma” is missing.

Page 2, material and methods section , “In order to remove the fibroblasts the cells suspension was pre-adhered in an uncoated 10 cm cell culture vessel for 30 min at 37°C”, “coma” is missing.

Everywhere for “m” symbol , “u” is used.

Discussion paragraph three, 2nd last line, “This studie”

Discussion 4th papargraph, page 10, 1st line, “significant-ly”

Author Response

Response to Reviewer 1 Coments

In materials and methods author mentioned that the astrocytes cultures were passaged for the first time, so when time course study performed, it was after passaged , can author provide immunostaining data of Day1 of culture, before passaging them.

Response to Reviewer 1 Comments (1): the cells were cultured in flasks for the first time before passaging. But, only astrocytes cultured on coverlids (after passaging) could be fixed, permeabilized, incubated with antibodies, etc.  This means, I could provide immunostaining data of astrocytes after passaging. But, the experiment with astrocytes in flasks can not be carried out.

Results section, page3, first paragraph, author mentioned in manuscript that “the majority of Calbindin D28k glial cells are stained positive for GFAP at day 1, while after 9 days only a small percentage of Calbindin D28k glial cells are stained positive for GFAP (Figure 1, Figure 2)” , here figure 1 shows day 5 immunostaining data not day 9 data, can author provide immunostaining data of day 9. (in earlier version Figure 2 was showing day 5 data, any reason of changing figure numbers?) 

Response to Reviewer 1 Comments (2): thank you for your feedback. I would like complete immunostaining of day 9. I would also like change figure numbers again, just like the last time. ( day 5 data, astrocytes with positive GFAP and positive Calbindin D28k. I think, maybe this is even more important for readers. This is why I changed the figure numbers).

Morphology of astrocytes look totally different in cultures of serum –free medium at day 5 (Figure 3), can author provide immunostaining data of culture at 24h in serum-free medium to make comparison.

Response to Reviewer 1 Comments (3): thank you for your feedback again. I would like complete immunostaining of day 1 (24h in serum-free medium) in order to compare.

For Figure 1, can author provide single stained (green and red channel) images as with merged image shown in figure 1, I only see two cells positive with Calbindin D28, and those are not GFAP positive  as claimed in results (1st paragraph)

 Response to Reviewer 1 Comments (4):  Figure 1, images from the confocal microscopy. These Images were taken with 3 colour channels at the same time. These are NOT merged images.  These are original images with green channel (pos. Calbindin D28k), red channel (pos. GFAP) and blue channel (nucleus). Nevertheless, you can see well 2 Calbindin-D28k positive and GFAP negative cells (green), many GFAP positive and Calbindin-D28k negative cells (red).

For figure 3,4, 5, 6, 7 and 8, can author provide merged and phase contrast images ?

Response to Reviewer 1 Comments (5): Figure 3, 4, 5, 6, 7 and 8, images from Epifluorescence microscopy. These Images were taken with 2 colour channels separately.  Of course, I could use computer programs to make merged images. But, I think, it is better to see the change of GFAP (red channel) with images with 2 colour channels separately. The phase contrast images are not suitable her. The goal is about the character of GFAP in calbindin-D28k immunoreactive astrocytes. With the phase contrast images, GFAP and calbindin-D28k immunoreactive astrocytes could not be detected.

Discussion, paragraph 3, author didn’t do any analysis on GFAP expression in 10% FBS cultures, can author provide image analysis or western blot data to claim that GFAP is downregulating after passage?

Response to Reviewer 1 Comments (6):  Discussion, paragraph 3, I have done the analysis on GFAP expression in 10 % FBS cultures:  see below.

“We observe that the expression of GFAP in calbindin-D28k immunoreactive astrocytes depends on conditions in the astrocytes' environment. We find that the presence vs. absence of FBS in the medium plays an important role. In a medium, which contains 10 % FBS, GFAP gets downregulated after passage.

With the computer-based calculation all calbindin-D28k astrocytes are presumably GFAP positive at the beginning of a passage in a medium with 10 % FBS.”

The goal is about the character of GFAP in calbindin-D28k immunoreactive astrocytes. The cell co-culture is carried out in this experiment. This means, there are also many calbindin-D28k negative astrocytes, but GFAP is positive. Western blot is not suitable her. The GFAP in total all astrocytes is detectable, but NOT in Calbindin-D28k immunoreactive astrocytes.

I have already showed images Figure 1 (day 1), Figure 2 (day 5) , Figure 11 (day 1), Figure 12 (day 5) (in 10 % FBS). I would like also show the image of day 9 in 10 % FBS. The number of calbindin-D28k positive glial cells on every coverlid was counted in this experiment. The percentage of calbindin-D28k and GFAP positive glial cells was calculated by dividing the number of positive cells by the total number of calbindin-D28k glial cells per coverlid. Every counting was repeated three times. I could not show the down-regulation of GFAP only with images.

Minor comments:

Examples of typos and grammar mistakes

Abstract, line 6, “differ-ently”

Abstract, line 7, “nov-elty”

Abstract, line 2nd to last , as-trocytes

Abstract, line 2, “Nevertheless, it known”, please correct the grammar.

Page 2, material and methods section, 2nd paragraph, line 3, “ 20 ul 1 mg/ml DNAse”, “coma” is missing.

Page 2, material and methods section , “In order to remove the fibroblasts the cells suspension was pre-adhered in an uncoated 10 cm cell culture vessel for 30 min at 37°C”, “coma” is missing.

Response to Reviewer 1 Comments (7): thank you for your feedback again. I have already corrected.

Everywhere for “m” symbol , “u” is used.

Response to Reviewer 1 Comments (8):

Two tissue flasks were coated with 10ug/ml of poly-L-Lysine overnight at 4°C. It is really right.

in a Hanks-balanced saline solution (HBSS) in which 2 ml contain 200 ul, 0.25 % trypsin and 20 ul, 1 mg/ml DNAse. It is really right.

The cells (20,000 cells/50 ul) were plated with poly-L-Lysine coated coverlids for histochemical staining. It is really right.

Discussion paragraph three, 2nd last line, “This studie”

Discussion 4th papargraph, page 10, 1st line, “significant-ly”

Response to Reviewer 1 Comments (9): thank you for your feedback again. I have already corrected.

Reviewer 2 Report

I am satisfied with the amendments and corrections, as well as with the level of discussion that improved the quality of the manuscript.

Author Response

 I am satisfied with the amendments and corrections, as well as with the level of discussion that improved the quality of the manuscript.

Response to Reviewer 2 Comments: Thany you for your message.